# Connecting Status and Professional Learning: An Analysis of Midwives Career Using the Place© Model

## Patricia Gillen [1,2]

1   Southern Health and Social Care Trust, 10 Moyallen Road, Portadown BT63 5JX, UK;
    patricia.gillen@southerntrust.hscni.net
2   School of Nursing, Ulster University, Shore Road, Newtownabbey BT37 0QB, UK

**Abstract:** This paper seeks to deconstruct the place of midwives as professionals using the novel interdisciplinary lens of the Place Model—an innovative analytical device which originated in education and has been previously applied to both teachers and teacher educators. The Place Model allows us to map the metaphorical professional landscape of the midwife and to consider how and where midwives are located in the combined context of two senses of place: in the sociological sense of public esteem and also the humanistic geography tradition of place as a cumulative process of professional learning. A range of exemplars will bring this map to life uncovering both the dystopias and potentially utopian places in which midwives find their various professional places in the world. The Model can be used to help student midwives to consider and take charge of their learning and status trajectories within the profession.

**Keywords:** midwife; profession; learning; esteem; Place; interdisciplinary

## 1. Introduction

Midwives have a wide ranging and uniquely skilled place in caring for women not only throughout pregnancy and childbirth, but also in antenatal and postnatal care; neonatal care; sexual health and fertility services in partnership with women and their families [1]. The esteem of midwives and their educational trajectories are matters which are both important and contested places. It is for this reason, that Clarke's [2] Place Model, which combines the sociological sense of place as status and the geographical sense of place as a position on a career long learning journey, can provide a useful combined set of lenses with which to view this unique profession.

Midwives are often the lead professional but also work in collaboration and partnership with women, their families and a diverse multidisciplinary team including Obstetricians, Allied Health Professionals and Social Care colleagues. Globally, there is growing recognition that in order to optimize outcomes, *all pregnant women need a midwife and some need a doctor* [3] (p. 323) and that there are improved outcomes for mothers and babies when care is provided by midwives who are *educated, trained, licensed, and regulated* [4] (p. 1). A position statement by the International Confederation of Midwives (ICM) (an accredited non-governmental organization) highlights that while midwifery is recognized as an autonomous profession in many countries that it is not yet afforded this status globally [5]. The ICM identified five key elements of an autonomous profession as: a unique body of knowledge; a code of ethics; self-governance; processes for decision-making by its members and recognition from society through regulation. Others have previously ascribed similar professional attributes such as: an expert esoteric body of knowledge which is profession specific; autonomy that helps to set the parameters for a discrete area of practice, thereby assigning social power to the

profession and ethical considerations [6]. However, more recently, Mivšek et al. [7] have highlighted three additional characteristics of professions: interprofessional collaboration, partnership with user and reflective practice. The ICM advocates for all countries to support midwives to promote midwifery as an autonomous profession, in order to optimize the care that they can provide for women and their families.

Midwifery often receives attention within the media not only about the outcomes of real-life maternity care but also portrayals of midwives in television dramas such as *Call the Midwife* which reflect, through somewhat rose-tinted glasses, on midwifery in the 1960's in an area of relative socioeconomic deprivation. More up to date documentaries such as *One Born every Minute*, while also based on reality, tend to focus on the 'best bits' of childbirth. This may have had the impact of attracting more applications to midwifery courses [8] but has done little to reflect the reality of a twenty first century midwife's role. This paper aims to use the Place Model to provide a more realistic and complex map of the profession. Clarke provides a usefully candid appraisal of the Model in relation to her maps of the teaching profession,

> *Whist undeniably reductionist in nature (like many models), the Place Model presents a usefully uncluttered landscape which is mapped in a way that is intentionally schematic rather than mathematical in nature (although it does look like a graph), a heuristic rather than a positivist equation. Like all maps, it is subjective, like all models it is wrong. Nevertheless, the Place Model is a map with a purpose. It is proffered as an interdisciplinary thinking tool for two key user groups: student professionals and their tutors. In preparing their students for their professional futures, tutors may invite them to consider critically their future learning journeys and status, across its terrain.* Clarke [9] (p. 73)

There are many, conceptualizations, many models, of professionalism. The Place Model focuses on a unique interdisciplinary combination of two senses of place (place as esteem) and place as location (i.e., akin to Massey's notion of Geographical imagination [10]), in this case, location along a career long learning journey. As students begin on this journey, it is useful for them to consider these two key aspects of professionalism-trustworthiness and expertise, especially at a time when trust in professionals is under attack in the public sphere. The model affords an opportunity to examine a range of dystopian aspects (and examples of these—which are not intended to be exhaustive) as well as pointing towards ideals.

It would appear that Midwives are recognized for the key role that they play in the health and well-being not only of childbearing women and their babies but also future generations. They seem to be valued and invaluable exemplars of trustworthy experts. However, all may not be as it seems. This paper will examine not only the place of the midwife from the perspective of their expertise and professional learning but also that of the place within public esteem, focusing mostly on the UK but also drawing on some key contrasting and comparable international examples.

## 2. Background

Documentary evidence of the role of Midwives can be found in early history and in a number of places in the bible with textbooks and training for midwifery being traced back to 17th century. Donnison [11] and Borsay and Hunter [12] provide useful accounts of the checkered history of the professionalization of midwifery, not least its' evolution from a 'female mystery' and wisdom [11] (p. 11) to one which has often become dominated by a male view of the world and the perceived need to control childbirth. The International Confederation of Midwives set global standards for midwifery practice, education and regulation [12–14]. These standards are reflected in midwifery education, practice and regulatory frameworks that are found in many countries. However, to enable global autonomous midwifery practice, it is expected that these standards provide not only a basis for the review of existing regulatory frameworks but also the impetus, guidance and direction to countries where regulatory frameworks for midwifery are limited or absent.

The International Confederation of Midwives (ICM) define a midwife as:

*'a person who has successfully completed a midwifery education programme that is based on the ICM Essential Competencies for Basic Midwifery Practice and the framework of the ICM Global Standards for Midwifery Education and is recognized in the country where it is located; who has acquired the requisite qualifications to be registered and/or legally licensed to practice midwifery and use the title 'midwife'; and who demonstrates competency in the practice of midwifery.* [1]

Across the world, country specific legal and professional structures are in place. While legislation is largely viewed as supportive of the profession, fear of litigation has been shown to have a negative impact on how midwives practice [15,16] with regulation and a blame culture and 'the fear factor of risk' [17] and litigation [18,19] inserting a very real fear factor within the realm of professional autonomy and judgement.In the United Kingdom (UK), for example, Midwifery is a protected legal function making it a criminal offence for anyone other than a registered midwife or medical practitioner, (except while in training or in an emergency) to attend a woman in childbirth [20]. All women have the right to access the care of a midwife free at the point of delivery and theoretically, taking account of their medical, childbirth histories and preferences can choose to deliver at home, in a birth center (Midwifery Unit) or in an Obstetric Hospital. There are also a number of independent midwifery care providers; some requiring payment, and some not as they are part of commissioned maternity care provision. However, there is evidence that some midwives, particularly those caring for and supporting women making unconventional birth choices are practicing in fear of litigation [21].

Midwives in the UK, as in many other countries, are a graduate profession and as such are regulated by the Nursing and Midwifery Council (NMC) which is turn is overseen by the Professional Standards Authority for Health and Social Care. At the point of registration, midwives are expected to have the necessary *'behaviors, knowledge and skills required to provide safe, effective, person-centered care and services'* [22] (p. 7). Underpinning these behaviors, knowledge and skills is professionalism, defined by the NMC as being

*' . . . characterized by the autonomous evidence-based decision making by members of an occupation who share the same values and education . . . '* [22] (p. 6)

A midwife's professionalism is demonstrated through being accountable, a leader, an advocate and being competent [23]. While it is postulated that Midwives are autonomous practitioners, it is clear that they work within a strict country specific legal, regulatory, professional and moral code of practice [24], are expected to follow best practice evidence based guidelines such as those of the National Institute for Health and Care Excellence (NICE) and work within their employer guidelines and policies. While most often the lead professional for women with a straightforward pregnancy, when necessary, close collaboration with obstetric colleagues in particular and other multidisciplinary colleagues, in partnership with women and their families is required to achieve optimal outcomes. Continuing professional development is not only an aspiration for midwives but in many countries a regulatory requirement in order to remain as a registered midwife and to continue to practice [24,25].

Midwives might also be seen to have a relatively strong global identity which is in part due to a network of midwifery associations across the world providing a sense of unity and support among midwives but also fostering strong relationships with policy makers and other health care professionals [25], potentially influencing health care policy and resource allocation. However, Castro Lopes et al. [26] highlight that being a predominantly female profession, gender issues and public opinion in some countries may negatively impact on midwives associations' relationships with those in authority and leadership, restricting their inclusion in key negotiations and discussions. Midwives have a long history of challenging on gender issues, not least in professional discussions where medicalized terminology such as diagnosis of pregnancy, symptoms of pregnancy and the notion of a pregnant woman's return to the normal state after birth [27], were often used by a predominantly male medical profession, until relatively recent times.

So, how can this Place model be applied within the context of midwifery to examine the status and professional learning of midwives?

## 3. Origins and Components of the Place Model

The origins of the model and how it evolved are explained in depth in Clarke [2], but in essence, in its original version, the model uses two perceptions of 'place' as the lens through which to examine the place of a teacher; (1) *'the humanistic geography tradition as a process-the career long professional learning journey'* and (2) *'place, in the sociological sense of teacher status'* [2] (p. 69). The combination of status and professional learning were recognized as key strengths of teacher education, and it was considered that this would be a useful lens through which to view midwives and explore their status and professional learning.

The subheading of the model is an important starting point for understanding the structure and application of the model within the context of midwifery. Replacing the original 'Who is teaching me today?' with' who is my midwife today? immediately puts the woman at front and center and places the midwife 'with woman'; the literal meaning of midwife.

The Place© model itself resembles a graph. The horizontal axis is a continuum: a cumulative, career-long, professional learning journey for the midwife (not a time-scale). It draws on Hoyle's [28] appraisal of professionality- where 'restricted' focuses on the individual's practice and autonomy and the 'extended' on wider society, multidisciplinary collegiality and career long learning and development towards being and becoming Clarke's (2019) [29] trustworthy experts. The vertical axis focuses on the status of the midwife, based on public perceptions of the esteem in which midwives are held, ranging from low to high. Clearly, midwives have less agency in relation to this but by seeking to control their learning trajectories they may seek to influence this key dimension of professionalism. As Figure 1 illustrates, the intersection of these axes affords the creation of four quadrants: proto-professionals, precarious professionals, professionalized and professional. A fifth element of the model sits outside the axes, where the answer to the question 'Who is my midwife today?' is 'no-one'; the reality for some women globally, particularly in low and middle income countries.

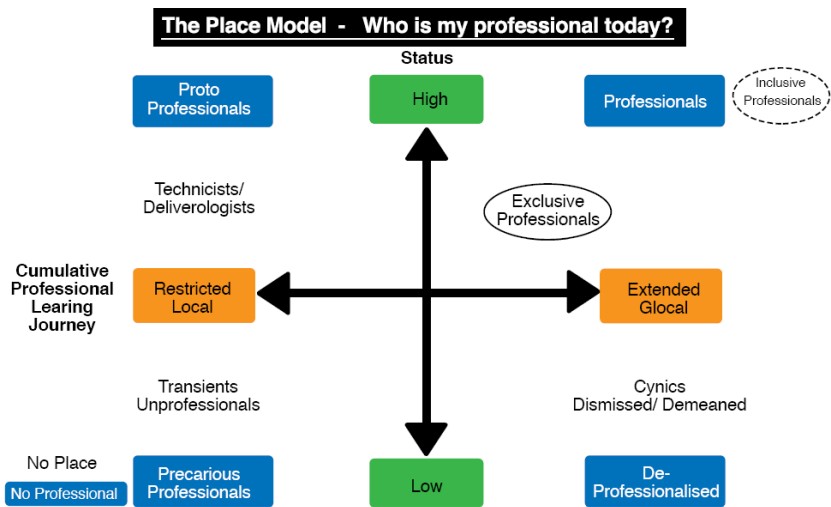

**Figure 1.** The Place Model.

## 4. The Place Model within the Context of Midwifery

Examples from midwifery will now be used to convert the Model to a 'living graph' (Leat, 1998) [30] in which we might consider how the Place Model can be used to *'map both career-long professional learning trajectories and to inform comparisons at individual and systemic levels'* [2] (p. 73) in a similar way to how it was used to explore teacher professionalism and status [2]. The starting point is the axis of the model, starting with the top left corner, the 'proto-professionals' before going off-axis

to the 'no midwife' component and then moving anticlockwise around the three remaining in-axis quadrants of the model, ending at the 'professionals' quadrant.

It is possible to 'populate' each of the five sections of the model's metaphorical landscape as a Living Graph (Leat, 1998) [30] using illustrative examples drawn from the profession of midwifery to bring the model to life and raise key questions about the profession.

### 4.1. Proto-Professionals

At the start point of the model- the top left-hand corner, the proto-professional is located. The prefix Proto is derived from Greek (*prôtos*) and means first and from pro meaning before [9]. Within the context of midwifery, this relates to the midwife as a newly registered and/or licensed practitioner. As the entry point for the profession, this quadrant allows us to explore the challenges faced by new graduates and registrants in order to help them to move along the continuum. However, it also affords us the opportunity to consider the relatively new attainment of a professional status for midwifery, the journey which began in the nineteenth century and despite strong medical opposition, was finally legislated for in the first decades of the twentieth century. It may be argued that the progression to a profession has led to the protection of the midwife's role in legislation but closer inspection may indicate that this may have come at the price of being less autonomous as a profession and more closely influenced by 'medical men' [11].

A midwife although signed off as competent and of 'good health' and character (considered to be capable of practicing safely and effectively [31]), they are still on a learning trajectory, will continue to develop professionally and given time and support, will continue to hone the recently acquired knowledge and skills from their education program. Many years ago, newly qualified nurses unpreparedness was described as 'a reality shock' [32] with Fenwick et al. [33] concurring with this and highlighting the importance of context and culture on the transition for new midwives and the need for strong relationships with midwifery colleagues to help them develop and grow in confidence. A structured and tailored induction and preceptorship (a period of support to help new registrants transition from student to registrant advocated by the NMC [34]) can enhance confidence as often a new registrant's experience is unstructured and insufficient to meet their needs [35].

The ICM set a benchmark for the standardization of midwifery education with midwifery being a graduate profession in many countries [14]. It also advocates that as it is an ethical duty for all midwives to provide safe practice that continuing professional development should be compulsory for all practicing midwives [25]. For some midwives, it may be argued that, the ongoing acquisition of knowledge and skills may be solely to meet their regulators' requirements or a means of staying on the professional register. However, for many midwives (one would hope most), the continual learning is driven by a passion to provide women and babies with the optimum quality of midwifery care. This requires not only a theoretical knowledge and understanding of the evidence base and its application to practice but also the acquisition of practical skills including high level communication skills in order to care for women and babies and support women and their partner (if appropriate) in their decision-making and choices.

As with other healthcare professionals, it can be difficult for midwives to find time to undertake these continuing professional development activities. While advocated by the ICM [25] and some countries regulators such as in the UK [36], the plethora of mandatory training for Health and Safety and other corporate priorities, coupled with staffing challenges, make it difficult for midwives to be released to undertake professional development activity. Often activity is undertaken in own time, sometimes at one's own expense with a promise of time in lieu or additional payment.

### 4.2. No Midwife

The Place Model provides the opportunity to consider the reality that 'no midwife' can present. The World Health Organization (WHO), recognize that within the right context, midwives who meet the ICM standards for education and regulation, can provide most of the fundamental care that is

needed by women and their babies [13,14]. However, in many countries in the world, women do not have access to a professionally trained midwife or skilled birth attendant [37]. The State of the World's Midwifery report [38] has highlighted that only four of the 73 middle-or low-income countries surveyed had midwives who were fully trained and skilled to provide care for women and their babies with fewer than half of these countries having legislation acknowledging midwifery as an independent profession [26]. A shortage of midwives is a worldwide phenomenon, including in high income countries where standardized and regulated midwifery education is available [39], with recruitment and more often retention of registered midwives, a key challenge maternity care providers grapple with for a multiplicity of reasons including workforce planning [40,41], work related stress [42] and bullying [43]. Failure to reach safe staffing levels, poor communication and professional collaboration has been shown to contribute to unsafe and substandard care [44].

In 2018, in recognition of the need for a definition of 'skilled health personnel', a joint statement was issued by the World Health Organization (WHO), the United Nations Population Fund (UNFPA), the United Nations Children's Fund (UNICEF), the International Confederation of Midwives (ICM), the International Council of Nurses (ICN), the International Federation of Gynecology and Obstetrics (FIGO) and the International Pediatric Association (IPA) [45]. One way through which to end preventable maternal mortality is to increase the number of births attended by skilled birth attendants or health personnel. Evidence of this is already clear, as during 2012–2017, almost 80 per cent of live births worldwide were attended by skilled health personnel, an increase of 62 per cent since 2000–2005 [46] with a decrease in the maternal mortality rate of 37% since 2000. However, further progress is needed if by 2030, Sustainable Development Goal 3 (SDG 3) to reduce the global maternal mortality ratio to less than 70 per 100,000 live births is to be achieved.

Conversely in high income countries, such as the UK, Australia and the US, there is an albeit small but growing number of women who choose to birth without a midwife or doctor present. Research by Jackson et al. [47] reports that women who free birth, do not view hospitals as safe places within which to birth. This view is further supported by other women who chose free birth in order to have choice, control and autonomy during birth [48] with some women relating the absence of the woman-centered care in maternity services, leaving them feeling vulnerable and unsafe [49]. It is important to note that safety relates not only to the physical sense of safety but also the emotional, psychological, and social aspects of care.

So clearly, while midwives are often recognized as skilled professionals, they are not always viewed as such and are seen to work within a system '*that can lead to a unique set of additional risks to the mother and baby*' [47] (p. 566). However, strong voices and evidence advocating midwifery led continuity of care models for most women [50,51] are leading to change. This will require not only the redesign of maternity care provision in many localities but also the education and willingness of midwives to work within continuity of care models. A continuity of care model is not a new phenomenon having been advocated for many years and one that exists in small pockets already. Indeed, it may be that midwifery is reverting back to a system of care delivery that prior to childbirth moving into the hospital setting and becoming more medicalized was the way in which care was provided.

## 4.3. Precarious Professionals

The next quadrant of the model challenges us to consider the Precarious professional. Precarious midwives are a particularly important group. Two contrasting professional trajectories are presented in this part of the model—both can produce damaging outcomes. Firstly, those who might be deemed to be unprofessional and secondly those who do not/cannot stay in the profession for long.

There are clear standards for the minimum education requirements expected for entry to a midwifery education program, however, it is much harder to make a judgement about the values and character of a potential student midwife [23]. Given the regulatory mechanisms within midwifery, and the global standards for education and practice, it would be expected that those deemed as unprofessional would be investigated through their education providers' or employers' disciplinary

system or by their regulator or both. In the UK, approximately 0.7% of midwives registered with the NMC were referred for Fitness to Practice concerns in 2017/18, accounting for 240 out of 35,830 registered midwives [52]. Referrals can be made by anyone including a service user, a member of the public, an employer or the police.

However, despite the great majority of women and their families, accessing the type and quality of care that they choose, in England for example, litigation in maternity care while only 10% of claims, accounted for 50% of the total value of claims [53]. Reports into failings in maternity services in the UK [54,55] have found that care provided by midwives having been subjected to scrutiny was substandard, that there was a lack of openness and honesty, were critical of midwifery supervision and of those who regulated and monitored the Trust. These failings are clearly distressing and life changing for women and their families and can impact on the confidence that women have in maternity services generally and midwives in particular. On a more positive note, a Cochrane review by Sandall et al. [50] concluded that midwife-led continuity models of care resulted in women being less likely to have interventions, with a greater chance of satisfaction with their maternity care and outcomes that were at least comparable with those of women who had accessed other models of care.

Despite clear standards for education and maternity care, globally, there is concern over the abusive and disrespectful practices perpetrated on women during childbirth [56]. A systematic review by Bohren et al. [57] identified a new typology of mistreatment of women during childbirth under seven themes: physical abuse, sexual abuse, verbal abuse, stigma and discrimination, failure to meet professional standards of care, poor rapport between women and providers, and health system conditions and constraints. It is clear that disrespect and abuse happens not only at an individual woman and precarious professional level but also at systems' level within health care organizations, despite an increasing body of evidence being accessible at both individual professional and organizational level. Custom and practice and cultural norms are sometimes used to justify the abuse that women are being subjected to during pregnancy and childbirth. Some women reporting that they are traumatized in childbirth not only by how they are treated but also through a lack of communication, control and consent [58].Continuing professional development is vital in ensuring that midwives maintain and update the necessary knowledge and skills which underpin respectful maternity care such as interpersonal skills, values and attitudes [59].

Midwives, who, for a variety of reasons, have both short careers and limited learning opportunities can have negative outcomes not only for the profession but also for the women and families for whom they provide care. Likewise, given the investment in their professional education of both the individual midwife and the taxpayer, it is important to understand what has contributed to the precarious status of these midwives. In the UK, Ball [42] reported that the top five reasons given for leaving the profession were: being unhappy with staffing levels; being dissatisfied with the quality of care they were able to give to woman and babies; being overworked; feeling unsupported by their manager; and being unhappy with working conditions. A recent paper by Harvie et al. [60] identified that Australian midwives are unhappy working within a fragmented system that did not allow them to provide care for women in the way they would like, with midwives under 40 years of age being particularly vulnerable. However, in contrast in Afghanistan [61], the primary reason for leaving was lack of security due to civil unrest and conflict, family disagreement, with increased workload without payment coming further down in the list of contributing factors. It is clear that governments and employers need to support their employees by addressing the specific issues that impact on midwives choosing to leave the profession.

In addition, it is important to consider the selection and recruitment practices used by academic institutions. It is easy to evidence if applicants meet the academic selection criteria but making a judgement about the values, motivation and strength of character needed to be a midwife; the ability to be 'heartstrong' [62] and compassionate is less straightforward.

*4.4. The-de-Professionalized*

Moving around the model to the next quadrant, the focus is on the de-professionalized. For midwives who are educated to the global standards set by the ICM [14], it would be expected that they would be professional at the point of registration and for that positive professional learning trajectory to continue throughout their professional working life [25]. This is often a far from straight forward trajectory and there are a range of circumstances which may lead to deprofessionalization, even for the most experienced midwives—some deprofessionalization is a product of intrinsic factors such as personal disposition, some is produced by a range of external factors.

De-professionalized midwives may be experienced midwives who for some reason are disillusioned and who are sometimes also discouraging and unsupportive to student midwives and new registrants. They are the midwives that the students least want as their practice mentors or new registrants don't want as their preceptors as they have little or no interest in teaching them or helping them to gain the experience, they need to develop their midwifery skills and competencies [35]. Reasons for deprofessionalization, may arise from the demands placed on midwives from the emotional and physical aspects of being a midwife. Some midwives can retire in mid to late 50's but as most midwives are women and may have worked part-time; they may have limited pensions so that it is not always possible for midwives to stop working when they would like to or when the job becomes too physically or emotionally demanding. A recent study commissioned by the Royal College of midwives highlighted that many midwives

> *felt exhausted by their day-to-day work, emotionally and physically drained, dreaded the thought of another day's work and seriously wondered how much longer they could carry on.* Hunter (2018) (p. 15) [63]

In addition, as mentioned previously, midwives practice within an increasing litigious environment and are fearful of making a mistake or missing something that may cause harm to a mother or baby [16,17,64]. Examples of externally driven deprofessionalization include overseas midwives who on travelling to UK are not assessed as competent by NMC Competence Centers or are assessed as not reaching the required competency standard in the English language. They may feel deprofessionalized and may actually be unable to join the profession in their new country of residence, at least temporarily until or if they meet the standards. However, given the shortages of midwives globally and the drive in many countries to recruit midwives from other countries, maternity care providers often offer additional support to these potential new registrants in order to assist with their transition. From the perspective of the healthcare provider, this may mean that overseas midwives are more closely supported during their initial months of experience, until they have proved themselves as safe practitioners. Despite the importance of and the increasing number of overseas midwives being employed across the world, Ohr et al. [65] are one of a few papers who report on the learning from the development and operationalization of a program to enhance the transition of overseas qualified nurses and midwives (OQNMs) to Australia. The program included cultural acclimatization and tailored support from leaders across the organization.

Crucially, some midwives within the deprofessionalized quadrant present a risk to the profession, organization and more importantly to the women and babies in their care. It is therefore vital that midwives that are identified as deprofessionalized, perhaps through their employers' appraisals system, are supported to identify how this can be addressed, most often through individualized continuous professional development which can be tailored to meet their needs or being supported to upskill in a particular aspect of midwifery care.

*4.5. The Professionals*

The final quadrant examines the high status, highly professional midwife in the light of regulation, guidance and professional expectations for continuous professional development, including research-based practice and critical reflection on practice. The professional midwives continue to grow,

enjoy the pursuit of learning, retaining and learning new skills and knowledge in order to meet the needs of the women and families they care for.

The day-to-day role model portrayed by the most professional midwives (Clarke's most *trustworthy experts*) is, perhaps, both unrecognized and poorly rewarded. It is useful however, for student midwives to consider who their midwife role models are and to plan their own learning trajectories towards careers which are built on the trustworthy expertise, which is needed in the profession, even whilst avoiding the more dystopian corners of the Place Model.

The great majority of women who access maternity services are satisfied with their care and outcomes. Renfrew et al.'s [4] evidence informed framework for maternal and newborn care has identified, as many as 56 outcomes for mothers and babies that could be improved by care that is within the scope of a midwives practice. It is clear therefore that where the professional midwife or 'skilled health personnel' provides care for women and their families that they have improved outcomes. Conversely, poor care may lead not only to death but also to morbidity that has long lasting negative impacts on the physical and psychosocial health and well-being of the woman and her family [66] leading to potential intergenerational health inequalities. Professional midwives don't only need to meet the core competencies [25] identified by ICM but also to have the ability to be continually adapt to the ever changing landscape of midwifery care; discerning about the evidence base and increasingly adept at the appropriate use of technology to support their practice [67,68].

In many countries, professional midwives are graduates, with increasing numbers of midwives achieving master's degree level education and being awarded a PhD or Professional Doctorate. However, having achieved these academic qualifications, there is limited structures to support them to continue in clinical practice while fully utilizing these higher-level skills for the good of their profession, women and their families. Often, the PhD midwives return to a similar role to the one they left in practice or move to a university setting, becoming subsumed into an academic life that is focused on research outcomes such as securing funding and peer reviewed publications. However, there are examples of structures that have been put in place to support clinical academic researchers to flourish in the recognition that '*A rich and diverse health research environment helps patients and invigorates the workplace*' [69] (p. 4). No one approach or model fits all and it is incumbent on the midwifery profession to find ways in which to provide the necessary infrastructure and support to help midwives within all types of roles and levels of expertise to thrive in order that women and their families can in turn be provided with optimum care.

## 5. Conclusions

The Place© model has helped us to consider how and where midwives are located: in the sociological sense of the status and also place as a cumulative process of professional learning, a widening horizon which combines learning in and from the local and the global. For student midwives in particular, this provides a unique interdisciplinary lens through which to consider not only the beginning of their professional journey but also how their status and continuing professional learning might evolve, including ways that are less than professional. Other models of professionalism do not map the dystopian pitfalls in ways which encourage them to proactively shape their learning trajectories. The status of midwifery is important not for status sake itself but rather for the opportunity that status and regard afford the profession to influence healthcare policy, drive optimum standards for education and practice, and further develop the evidence base through high quality interdisciplinary collaborative research. The need for the recognition and support for midwives to have continuing professional development opportunities in order to best meet the needs of the women and babies they care for is clear. However, while there is evidence to support the assertion that care provided by professional midwives leads to improved outcomes for women and babies [4], it is also known that some women have no access to midwives (often not by choice), there are midwives who are precarious professionals, career-long proto-professionals or those midwives who have become de-professionalized. The reasons for midwives remaining as proto-professionals and becoming de-professionalized are

varied but given the need to provide women, her baby and family with the optimum quality of care and the ongoing challenges with recruitment, there is a need to find way to shift all midwives towards the high status, high learned professional. For those women who have 'no midwife', WHO [70] recognizes the need for "Competent, motivated human resources" as one of the eight domains of the WHO framework for the quality of maternal and newborn health care. Globally, midwives need to continue to work closely with women and to use the evidence base that demonstrates their effectiveness in the provision of high-quality maternity care and outcomes to influence and encourage interprofessional collaboration across the increasingly complex topography of multi professional and agency maternity care.

In the UK, the NMC is currently consulting on new education standards and proficiencies for the Future Midwife, looking towards 2030 in order to forward plan education that will prepare midwives to meet the needs of the woman, baby and families of the next decade and beyond. This is a challenge as the diversity and complexity of the childbearing women needs to be balanced with the normal physiology of childbirth. In all of this, it is important to consider what matters to midwives. How best can the profession of midwifery continue to attract bright, articulate and 'heartstrong' individuals to not only become a compassionate, skilled and professional midwife but to continue to be one throughout their professional career. We know that midwives value the support of their managers and colleagues as a way of minimizing the fear factor and maximizing their potential to support women to achieve optimal outcomes [71]. After all, uniquely amongst the professions, the most fundamental place of midwives is enshrined in the name—midwife meaning 'with woman' and the Place Model© has moved beyond this intimate scale to permit a broader mapping with regard to status and professional learning both nationally and internationally.

**Funding:** This research received no external funding.

**Conflicts of Interest:** No potential conflict of interest was reported by the author.

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
