# Peer review of "Connecting Status and Professional Learning: An Analysis of Midwives Career Using the Place© Model"

_education, doi:10.3390/educsci9040256_

Round 1

Reviewer 1 Report

The PLACE model does not answer the questions posed by the researchers.   In the section Origins and components of the Place Model should explain better in what this methodology consists.    The bibliographical references are in different citation models.

Author Response

Thank you for your comments, please see my response in the table below. The paper with changes highlighted in yellow is attached.

Reviewer 1 Place Model paper Midwifery

Question

Comment

Response

Does the introduction provide sufficient background and include all   relevant references?

Must be improved

The title and abstract have been revised and the introduction has   been further developed. All   changes to the paper in response to both reviewers’ comments are highlighted   in yellow.

Is the research design appropriate?

Must be improved

It is now stated in the abstract that this is an analytical rather   than empirical paper. Also further detail about the Place Model has now been   included under the Origin and Components heading.

Are the methods adequately described?

Not applicable

Not applicable

Are the results clearly presented?

Must be improved

The application of the PLACE model within the context of midwifery   has been further developed and explained.

Are the conclusions supported by the results

Not applicable

Not applicable

Comments and Suggestions for   authors

The PLACE model does not answer the questions posed by the researchers

The title of the paper has been amended.

In the section Origins and Components of the PLACE model should   explain better in what this methodology exists

Further explanation about the PLACE model is now included in this   section.

The bibliographical references are in different citation models

These have been corrected to ensure that all references now conform   to Education Sciences format.

Reviewer 2 Report

The paper has a number of concerning claims and omissions and at times has a patronizing tone.

Who is the audience for this work? It would not likely be midwives, their health professional colleagues or their employers who would know the qualifications and scope of practice of midwives.. Please comment.

The paper refers to the contested place of  midwives, contested by whom?

The focus of the paper drifts from perspectives of expertise and professional learning to place within public esteem, to  midwifery as a protected legal function, to no midwife, what is the purpose of the place model? Who will use the model and how?

The full scope of practice of the midwife does not seem to be understood by the authors, limited only to pregnancy and childbirth in this paper, this needs to be expanded to antenatal, postnatal care, care of the neonate, sexual health, fertility and supported by the literature.. 

The understanding of a normal state of a woman's body is also not understood by the authors- on page 3 there is reference to 'a pregnant woman's return to the normal state after birth'. Are the authors suggesting being pregnant is not normal?

Equally, what is meant by a midwife 'being competent and of good health and character'  P4?

What is meant by  'it can be difficult to find time to undertake development activities'? Are the authors suggesting midwives would find it more difficult than other health professions? Are the authors suggesting that only midwives attend these activities in their own time and at their own expense? these statements need to be supported with evidence from the literature.

How has the section of nearly a full page on 'No Midwife' including freebirthing, contributed to this paper?

P6 there is a paragraph referencing  abuse of women in pregnancy and childbirth without any clear link to how this relates to midwifery professional learning. This needs to be removed or a link needs to be made.

P7 what is the link with status and 'ideal type'?

P8 what is meant by an ideal midwife (high status, learned professional)  and also less ideal versions?

Why would authors think any health professional would think being a midwife is 'based on reality TV or dramatisation which tend to have a rose tinted hue"?

P9 What is meant by 'minimise the fear factor'? Who is fearful?

Author Response

Thank you for your comments. Please see my response in the table below and the paper with changes highlighted in yellow attached.

Place Model Paper Midwifery July2019

Reviewer 2

Question

Comment

Response

Does the introduction provide sufficient background and include all   relevant references?

Must be improved

The title and abstract have been revised. All changes to the paper in response to both   reviewers’ comments are highlighted in yellow.

Is the research design appropriate?

Must be improved

Further detail about the Place Model has now been included under the   Origin and Components heading.

Are the methods adequately described?

Must be improved

It is now stated in the abstract that this is an analytical rather   than empirical paper and further detail included about the PLACE model as   above.

Are the results clearly presented?

Must be improved

The application of the PLACE model within the context of midwifery   has been further developed and explained.

Are the conclusions supported by the results

Must be improved

The conclusion has been revised.

Comments and Suggestions for authors

The paper has a number of concerning claims and omissions and at   times a patronising tone

I have read the feedback carefully and believe that I have addressed   the ‘concerning claims and omissions’ as pointed out by the reviewer.  The author is a midwife since 1987 and   currently working in academia and in the practice setting undertaking   research and development activity with midwives and service users.

Who is the audience for this work? It would not likely be midwives,   their health professional colleagues or employers who would know the   qualification and scope of midwives. Please comment

The author believes that this paper would be of interest to midwives   working in practice, research and education.

The paper refers to the contested place of midwives, contested by   whom?

Contested has been removed in the editing of the abstract.

The focus of the paper drifts from perspectives of expertise and   professional learning to place within public esteem, to midwifery as a   protected legal function, to no midwife. What is the purpose of the place   model?  Who will use the model and how?

Further explanation is provided about the model generally and each   quadrant specifically.

The full scope of practice of the midwife does not seem to be   understood by the authors, limited only to childbirth and pregnancy in this   paper. This needs to be expanded to antenatal, postnatal care, care of the   neonate, sexual health, fertility and supported by the literature

First paragraph in paper revised to reflect the broader remit of the   midwife, supported by the literature.

The understanding of a normal state of a woman’s body is also not   understood by the authors- on page 3 there is a reference to ‘ a pregnant   woman’s return to normal state after birth’ are the authors suggesting that   being pregnant is not normal?

P3. This has been rephrased to make it clearer that terms such as ‘ a   pregnant woman’s return to normal state after birth’ have been used by   medical colleagues  and midwives have challenged   the use of this terminology for many years (Rothman 1982).

Equally, what is meant by a midwife’ being competent and of good   health and character? P.4.

This has now been clarified under Proto-professionals on page 5.

What is meant by ‘it can be difficult to undertake development   activities’? are the authors suggesting midwives would find it more difficult   than other health professionals? Are the authors suggesting that only   midwives attend these activities in their own time and at their own expense?   These statements need to be supported by evidence from the literature.

This has been further clarified to recognise that this is a challenge   faced by most healthcare professionals.

How has the section of nearly a full page on ‘No Midwife’ including   freebirthing contributed to this paper

The No Midwife relates to the second quadrant of the Place© model.   There is strong evidence that midwives positively impact on the health and   well-being of women and babies (the Lancet Series 2014). However, the   phenomenon of free birthing is growing in high income countries and is   indicative of a lack of women’s trust in maternity care services and fear of   unnecessary interventions  including   care provided by midwives.

P6. There is a paragraph referencing abuse of women in pregnancy and   childbirth without any clear link to how this relates to midwives   professional learning. This needs to be removed or a link needs to be made.

This has been revised to make the link with professional learning.

P7. What is the link with status and ‘ideal type’?

Status has been addressed throughout paper and ‘ideal’ removed

P8 What is meant by an ideal midwife (high status, learned   professional) and also less ideal versions?

Ideal has been removed, see page 10 under ‘the professionals’-with   links to regulation, education and practice.

Why would authors think any health professional would think being a   midwife is ‘based on reality TV or dramatization which tend to have a rose tinted   hue?

In the UK, there is some evidence that applications to the midwifery   profession have increased as a result of TV dramas such as ‘Call the Midwife’   (Tiran 2012). However, sentence relating to this on page 8 removed.

P9 What is meant by ‘minimise the fear factor? Who is fearful?

There is evidence that midwives are feeling vulnerable and fearful in   practice particularly when caring and supporting women who have made   unconventional birth choices (Feeley et al, 2019;Scammell, 2016,; Dahlen 2010).

Reviewer 3 Report

This paper could be an interesting contribution to the field but it has numerous problems with sentence flow, punctuation and most significantly, clarity.  It appears to be a survey of the literature, primarily in the UK, using a descriptive analysis model taken from education.  There is an assumption made by the authors, but not convincingly argued in the paper, that midwives can be viewed in terms of status and competency like "teachers".  I found the paper difficult to follow, not because the analytical model isn't described clearly, but because the arguments made by the authors regarding the profession of midwifery were presented as widely agreed upon conclusions and/or never fully explained to the reader.  While the paper is well cited, the discussion is not well articulated.  

More importantly, there is little critical analysis or discussion of how this analysis contributes to the field.  While looking through a new lens is interesting, there is little documentation of how/in what ways the analysis improves our general understading of the status of midwives or how the analysis could/should be used to improve education/training, research, practice or policy.  I am not sure what the point of the paper is and was confused by the inadequate explanation of the purpose for practice of this paper.  While the model is useful for 'organizing' the literature and what is known about the theory and practice of midwifery, I didn't learn anything new from the paper.

A revision/rewrite is necessary for the authors to clarify their point of the paper for readers and to refine their presentation so as to be a more comprehensible representation of their arguement.  Finally, the authors make conclusions about the literature they cite but do not really explain the points they base their conclusions on nor do they clearly present the reasoning for the conclusions they come to in their own paper.

Author Response

Reviewer 3 Place Model paper Midwifery

Question

Comment

Response

Does the introduction provide sufficient background and include all relevant references?

Can be improved

 This section has been revised

Is the research design appropriate?

Can be improved

This section has been revised

Are the methods adequately described?

Can be improved

This section has been revised

Are the results clearly presented?

Can be improved

This section has been revised

Are the conclusions supported by the results

Can be improved

This section has been revised

Comments and Suggestions for authors

This paper could be an interesting contribution to the field but it has numerous problems with sentence flow, punctuation and most significantly, clarity.  

Thanks to the reviewer 3 for their feedback.

All changes to the paper in response to the reviewer’s comments are in tracked changes.

Revisions made throughout the paper to address sentence flow, punctuation and most significantly, clarity.  

It appears to be a survey of the literature, primarily in the UK, using a descriptive analysis model taken from education.  There is an assumption made by the authors, but not convincingly argued in the paper, that midwives can be viewed in terms of status and competency like "teachers".  

Argument has been strengthened throughout paper ( see tracked changes in paper).

I found the paper difficult to follow, not because the analytical model isn't described clearly, but because the arguments made by the authors regarding the profession of midwifery were presented as widely agreed upon conclusions and/or never fully explained to the reader.  While the paper is well cited, the discussion is not well articulated.  

The conclusions about the midwifery profession are drawn from empirical research and regulatory and professional organisations. In the paper, I have strengthened the arguments within the confines of the word count.

More importantly, there is little critical analysis or discussion of how this analysis contributes to the field.  While looking through a new lens is interesting, there is little documentation of how/in what ways the analysis improves our general understanding of the status of midwives or how the analysis could/should be used to improve education/training, research, practice or policy.  

There is further discussion about how this analysis contributes to the field.

I am not sure what the point of the paper is and was confused by the inadequate explanation of the purpose for practice of this paper.  While the model is useful for 'organizing' the literature and what is known about the theory and practice of midwifery, I didn't learn anything new from the paper.

The point of the paper has been further explicated

A revision/rewrite is necessary for the authors to clarify their point of the paper for readers and to refine their presentation so as to be a more comprehensible representation of their argument.  Finally, the authors make conclusions about the literature they cite but do not really explain the points they base their conclusions on nor do they clearly present the reasoning for the conclusions they come to in their own paper.

 There has been a considerable revision of the paper to address the reviewers comments

Round 2

Reviewer 1 Report

The article has not improved despite the changes introduced.

It is not clear why the PLACE model should be used and not another, and how it has been validated for midwives.

It is necessary to provide more bibliography that encourages the use of the PLACE model for midwives.

Author Response

Thank you for your comments; my response is in italics.

It is not clear why the PLACE model should be used and not another, and how it has been validated for midwives.

This paper describes the first application of the PLACE model to midwifery. It has therefore not been validated for midwives but is a model that has previously been used to explore the  'metaphorical professional landscape' of teachers (Clarke 2016, p2), it is therefore a novel application of this model to another professional group. 

It is necessary to provide more bibliography that encourages the use of the PLACE model for midwives.

Throughout the paper, there are references to specific professional literature from both a regulatory and professional perspective. The author hopes that the application of the PLACE model to midwifery will encourage readers to consider the status and professional learning of midwives from a different perspective. As the PLACE model has not been used before in midwifery, it is not possible to provide further bibliography.

Reviewer 2 Report

The author has responded well to the suggestions and questions of the reviewer and the paper is stronger and now likely to be of interest to a wider audience. 

The paper would benefit from assistance with sentence structure where referring to the work of others, some sentences seem incomplete referring to the reference number mid sentence rather than the authors' names. e.g. Lines 185 and !89 P 5

Author Response

Thank you for your comments; my response is in italics.

The author has responded well to the suggestions and questions of the reviewer and the paper is stronger and now likely to be of interest to a wider audience. 

The author appreciates the positive comments from the reviewer in response to the changes made to the paper further to the previous constructive feedback.

The paper would benefit from assistance with sentence structure where referring to the work of others, some sentences seem incomplete referring to the reference number mid sentence rather than the authors' names. e.g. Lines 185 and 189 P 5

In addition to accessing the referencing guidelines, I have have asked for and taken direction from the Assistant Editor and have made corrections to references throughout the paper, (highlighted in yellow), where appropriate.

Reviewer 3 Report

I have commented extensively in the text. However my essential point is that the "so what" of the article, while improved, has still not been clearly articulated.  It appears that the authors are attempting a minor comparison of the Professional model analysis of the midwifery profession to the Place model. However, this point is never fully articulated or clearly presented.  It is not clear how the Place model improves on the Professionalism model or if, indeed, the Place model is in any way different to the professionalism model (e.g., Clarke which is cited) for an analysis of the context of midwifery in the world.

The problems/challenges for midwifery that are identified interestingly in the four quadrant Place model seem to me would be the same/similar if these points had been made using a Professionalism model of analysis.  It is not really clear how a reorganization of the challenges to midwifery into these four quadrants contributes anything new to the field.  Moreover, just reorganizing the existing problems doesn't seem to be any more advantageous for student understanding of the profession and its challenges.  

There is a glimmer of hope early in the conclusion that could be fleshed out but are not.  I feel the paper requires one more revision: clarify if you are indeed arguing against or for a professionalism model or be more explicit on how the four quadrants of the Place model contributes to a better/improved understanding of the field and/or offers a methodology to help students gear up for the challenges identified and how/what that improvement really is.  What is the 'so what' of this article?

Author Response

Thank you for your feedback. In particular, I have responded to:

'...clarify if you are indeed arguing against or for a professionalism model or be more explicit on how the four quadrants of the Place model contributes to a better/improved understanding of the field and/or offers a methodology to help students gear up for the challenges identified and how/what that improvement really is. What is the so what of this article?'

Please see the revised paper attached, with revisions in tracked changes on pages 2 & 10.

Round 3

Reviewer 1 Report

They have not made the recommendations of the previous review.

Author Response

I have spoken with Farrah Sun about this review